# The genome of pest *Rhynchophorus ferrugineus* reveals gene families important at the plant-beetle interface

Khaled Michel Hazzouri[1], Naganeeswaran Sudalaimuthuasari[1], Biduth Kundu[2], David Nelson[3], Mohammad Ali Al-Deeb[2], Alain Le Mansour[4], Johnston J. Spencer [5], Claude Desplan [3 ✉] & Khaled M. A. Amiri [1,2 ✉]

The red palm weevil, *Rhynchophorus ferrugineus*, infests palm plantations, leading to large financial losses and soil erosion. Pest-host interactions are poorly understood in *R. ferrugineus*, but the analysis of genetic diversity and pest origins will help advance efforts to eradicate this pest. We sequenced the genome of *R. ferrugineus* using a combination of paired-end Illumina sequencing (150 bp), Oxford Nanopore long reads, 10X Genomics and synteny analysis to produce an assembly with a scaffold N50 of ~60 Mb. Structural variations showed duplication of detoxifying and insecticide resistance genes (e.g., glutathione S-transferase, P450, Rdl). Furthermore, the evolution of gene families identified those under positive selection including one glycosyl hydrolase (GH16) gene family, which appears to result from horizontal gene transfer. This genome will be a valuable resource to understand insect evolution and behavior and to allow the genetic modification of key genes that will help control this pest.

[1] Khalifa Center for Genetic Engineering and Biotechnology, United Arab Emirates University, PO Box 15551, Al Ain, UAE. [2] Department of Biology, United Arab Emirates University, PO Box 15551, Al Ain, UAE. [3] Center for Genomics and Systems Biology, New York University Abu Dhabi, PO Box 129188, Abu Dhabi, UAE. [4] Date Palm Tissue Culture, United Arab Emirates University, PO Box 15551, Al Ain, UAE. [5] Department of Entomology, Texas A&M University, TAMU 2475, College Station, TX, USA. ✉email: cd38@nyu.edu; k.amiri@uaeu.ac.ae

The order Coleoptera is the largest among insects and has over 400,000 species, which account for more than 20% of metazoans[1] and include agriculture and forest pest species. Specialized interactions with host plants allowed their evolution as destructive herbivores and crop pests[2,3]. *Rhynchophorus ferrugineus* (*R. ferrugineus*) (Olivier 1790) is a Coleopteran pest in the Curculionoidea family whose larvae destroy palm trees worldwide. A native to Southeast Asia and Melanesia, its range has recently expanded due to accidental introductions into the Middle East, Mediterranean Basin, Caribbean, and USA[4]. It attacks more than 26 palm species belonging to 16 genera and has been classified as a serious pest on the A2 list according to the EPPO2008[5] (European and Mediterranean Plant Protection Organization). For instance, in 2009, the annual loss in the Arabian Gulf region, which accounts for 30% of the world date palm (*Phoenix dactylifera*) production, has been estimated at US$ 25.92 millions[6]. Symptoms of infestation are visible only after the tree has been severely damaged, thus destroying the tree beyond remediation before the pest is detected. This stealth lifestyle of the *R. ferrugineus* larva is enabled by its early migration to the heart of the date palm vascular system[7].

In general, dry woody plants have limited sugar, nutrient, and mineral content because of the lignified nature of plant cell walls. However, palm trees are wet woody plants that have a very sugary sap. Pest species of woody plants have to be detoxified from secondary metabolites such as allelochemicals, which requires metabolic adaptation[8,9] that also enables the pest to develop rapid metabolic resistance to other toxins, including insecticides. Indeed, multiple phytophagous beetles have increased activity of insecticide detoxifying enzymes, such as cytochrome P450s (CYPs), glutathione S-transferases (GSTs), and UDP-glycosyltransferases (UGTs)[10–13], as well as xenobiotic transporters[14,15]. They also have plant cell wall degrading enzymes (PCWDEs) for cellulose, hemicellulose, or pectin[16–18]. In fact, some beetles appear to have acquired PCWDEs via horizontal gene transfer (HGT) from fungi or bacteria followed by gene duplication and expansion into multigene families[19]. In contrast, other wood-feeding insects such as termites, ants, and cockroaches host microbial symbionts that provide these metabolic activities[20–23].

Despite recent investigations of gene expression in *R. ferrugineus*[24–26], no genome was available. To address this gap, we performed whole genome sequencing and de novo assembly of its genome, transcriptome sequencing, genome annotation, and performed comparative genomic analyses with the mountain pine beetle (*Dendroctonus ponderosae*), the coffee berry borer (*Hypothenemus hampei*), the red flour beetle (*Tribolium castaneum*), and *Drosophila melanogaster*. We report the evolution of gene families in *R. ferrugineus* and demonstrate the duplication and amplification of detoxifying genes and insecticide resistance genes (e.g., Rdl). We document the origin of gene families by HGT events, such as that of the glycosyl hydrolase (GH16). We also estimate ancestral and recent effective population size of the species and investigate whether there was selective pressure on some gene families involved in the adaptation to life on date palm tissue.

This *R. ferrugineus* genome will be an essential resource to study the genetic diversity of the species and will allow genetic manipulation via CRISPR/Cas9. Self-propagating of deleterious genetic variants could spread through the *R. ferrugineus* population through gene drives, which could weaken or eliminate the ability of *R. ferrugineus* to infest and destroy date palm plantations, thus saving millions of dollars and years of labor as well as maintaining a stable food supply for vulnerable communities.

## Results

### Genome assembly, characterization, and annotation

The genome of *R. ferrugineus* (Fig. 1a) is the third Coleoptera genome sequenced in the Curculionoidea family and a distant relative of model insect species *T. castaneum* (a Tenebrionid that diverged 234 Mya) and *D. melanogaster* (a Diptera that diverged 294 Mya) (Fig. 1b). They are distributed in South East Asia and in the Middle East (Fig. 1c).

The Supernova assembly was used to scaffold ABYSS assemblies from 150 bp Illumina paired-end data. This gave an improved assembly with N50 of 150.8 kb for female and 137.7 kb for male and an assembly size of ~780 Mb (male and female) (Table 1). Other methods were applied but did not yield significant improvements in assembly quality (Supplementary Fig. 1). The Nanopore long reads comprised 23 Gb of data with mean read length of ~2 kb (Supplementary Fig. 2) and an assembly of 474.4 Mb and a scaffold N50 of 79.8 kb. The hybrid Illumina paired-end (150 bp) and long-read Nanopore data generated an assembly of 789.9 Mb for female and 780.2 for male with an N50 ~2 Mb. Merging the two assemblies above did not improve the N50. A final chromosome-level assembly consisting of nine pseudochromosomes and an X chromosome was produced assuming syntenic relationships with the red flour beetle using Chromassemble (Fig. 2a). A summary of different assemblies generated in comparison to the red flour beetle is presented in Table 1. The final genome assembly of *R. ferrugineus* is deposited and available at NCBI and Dryad public databases (see "Data availability" section).

The *R. ferrugineus* karyotype comprises ten autosomes and a pair of sex chromosomes X and $y_p$[27] with males having X and $y_p$ and females XX. We identified 54 scaffolds that match the X chromosomes of the red flour beetle (Supplementary Fig. 3). The diversity of these scaffolds was low compared to the nine autosomes (Supplementary data file 1). Nucleotide diversity (pi) was $0.009 \pm 0.006$, on the X, whereas the autosomes had a diversity of $0.012 \pm 0.006$. However, the parachute $y_p$ scaffolds were hard to detect because of the degeneracy of this Y chromosome. We managed to identify scaffolds in the assembly that are likely part of the $y_p$ sex chromosome, however, more data will be needed such as a genetic map, in order to generate linkage groups and be able to orient anchored scaffolds on all of these pseudochromosomes and generate full oriented chromosomes.

The genome size of *R. ferrugineus* based on flow cytometry of two batches of individuals (five males or five females) was estimated to be between 696.3 and 726.2 Mbp (Supplementary Fig. 4). Kmer-based estimates of genome size were 626 for males and 603 Mb for females (Supplementary Fig. 5). These estimate are much larger than the genome size of *D. ponderosae* (257.091 Mb) and of *H. hampei* (151.27 Mb), which may be explained by greater transposable element content (see below). The *R. ferrugineus* genome has a G + C content of ~32%, which is similar to the other insect species[28]. We also assembled the mitochondrial genome that was 16,074 bp, with 38 genes typical of insect mitochondrial genomes, including 13 protein coding genes, two ribosomal RNAs (rRNA), 23 transfer RNAs (tRNA) and an A + T rich region (Supplementary Fig. 6, Supplementary data file 3).

### Annotation of the *R. ferrugineus* genome

We used the GeneMark prediction method to annotate the genome, resulting in 45,876 and 45,615 gene models in the two sequenced groups of individuals. Augustus de novo gene prediction gave rise to 27,119/27,116 gene models. EVM gene prediction used 149,007 insect transcripts and 114,218 insect protein models, resulting in 64,091/63,695 gene models. By combining all predictions, we annotated 25,567 good quality gene models. Non-coding tRNA gene prediction resulted in 1024 tRNA genes. Eighty percent of predicted proteins were found against the NCBI-Insecta

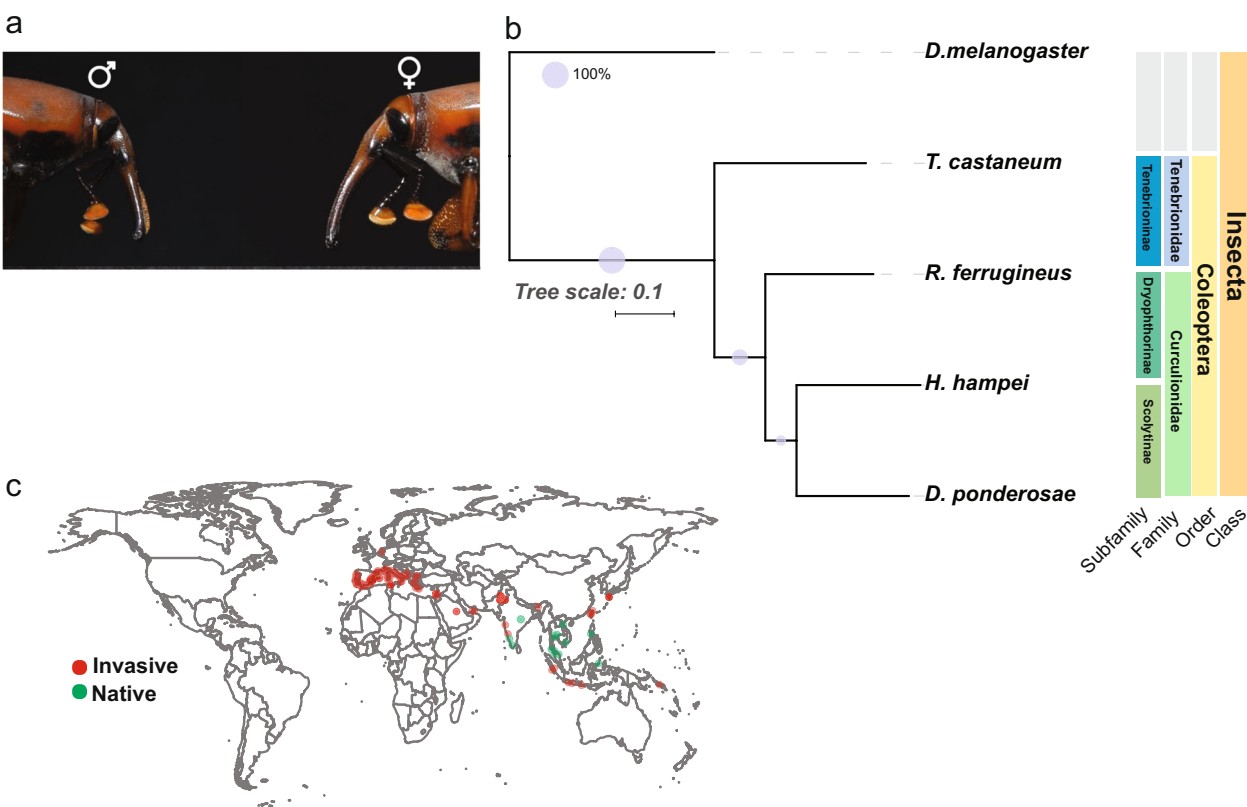

**Fig. 1 Geographic and phylogenetic context. a** *R. ferrugineus*, male and female. The male weevil has a tuff of soft reddish brown hairs along the dorsal facet of the snout, which is absent in the female weevil. **b** Phylogenetic tree depicting the relationship between *R. ferrugineus* and other Coleoptera from the Curculionidae and Tenebrionidae families with *Drosophila melanogaster* as outgroup. **c** Geographic distribution of the native and invasive *R. ferrugineus*[4] was plotted using R map package[89,100].

**Table 1 Assembly statistics of red palm weevil (*R. ferrugineus*) genome at different platforms in comparison to red flour beetle (*T. castaneum*).**

| Species | Data | Sex | Size (Mbp)* | Assembly size (Mbp) | Scaffolds** | N50ᵃ/ᵇ (Mbp) | BUSCO*** C (%) |
|---|---|---|---|---|---|---|---|
| *R. ferrugineus* (M_v.1) | Illumina paired-end + 10x genomics | M | 696.3 ± 5.3 | 780.5 | 12,462 | 0.1377/0.024 | 84.6 |
| *R. ferrugineus* (F_v.1) | Illumina paired-end + 10x genomics | F | 726.2 ± 12.8 | 783.3 | 12,355 | 0.1508/0.029 | 84.6 |
| *R. ferrugineus* (M_v.2) | Oxford Nanopore | M | 696.3 ± 5.3 | 474.4 | 10,580 | 0.0798/0.017 | 83.2 |
| *R. ferrugineus* (M_v.3) | Hybrid assembly (Illumina + Oxford Nanopore) | M | 696.3 ± 5.3 | 780.2 | 4822 | 2.12 | 89.2 |
| *R. ferrugineus* (F_v.3) | Hybrid assembly (Illumina + Oxford Nanopore) | F | 726.2 ± 12.8 | 789.9 | 4788 | 2.02 | 89.2 |
| *R. ferrugineus* (M_pseudochr) | Synteny to red flour beetle | M | 696.3 ± 5.3 | 782.19 | 4812 | 64.11 | 92.6 |
| *R. ferrugineus* (F_pseudochr) | Synteny to red flour beetle | F | 726.2 ± 12.8 | 780.66 | 4515 | 60.8 | 91.9 |
| *T. castaneum* (red flour beetle) | Sanger + BACs + Genetic + maps + Illumina + BioNano | M + F | 204 | 165.9 | 6580 | 14.6/0.073 | 98.4 |

*Flow cytometry estimation.
**Scaffold numbers.
***Benchmarking Universal Single-Copy Orthologs.
ᵃN50 scaffold.
ᵇN50 contig.

database while ~78% of proteins shared homology in the UniProt database. 8,726 (34%) proteins were annotated against the KEGG pathway database (Supplementary data files 4–8).

The quality of the initial assembly was assessed by comparing the genome against the BUSCO arthropoda database. Approximately 84.6% (1404/1658) of complete BUSCO gene models (567 Complete and single-copy and 837 Complete and duplicated) were annotated in the assembled male weevil genome. Approximately 12% of BUSCO gene models were found in a fragmented form and ~15% were not annotated in either genome. CEGMA analysis identified ~88.31% (219/248) of ultra-conserved Core Eukaryotic Genes (CEGs) in the genome. The

pseudochromosome assembly through synteny with *T. castaneum* showed improved N50 as well as BUSCO scores (Table 1).

The genome of *R. ferrugineus* has 25,567 genes. The average exon length was 230 bp with an average of 4.4 exons per gene (maximum: 56) while the average intron length was 681 bp. In contrast, the genome of *D. ponderosae* has 14,166 genes with an average exon length of 151 bp, an average of 8.3 exons per gene (maximum 85) and an average intron length of 1488 bp. *H. hampei* has 10,213 genes, with an average intron length of 1023 bp and an average exon length of 101 bp (6.3 exons per gene, maximum 62). *T. castaneum* has 14,467 genes, with an average exon length of 162 bp and an average intron length of 105 bp (Supplementary Fig. 7).

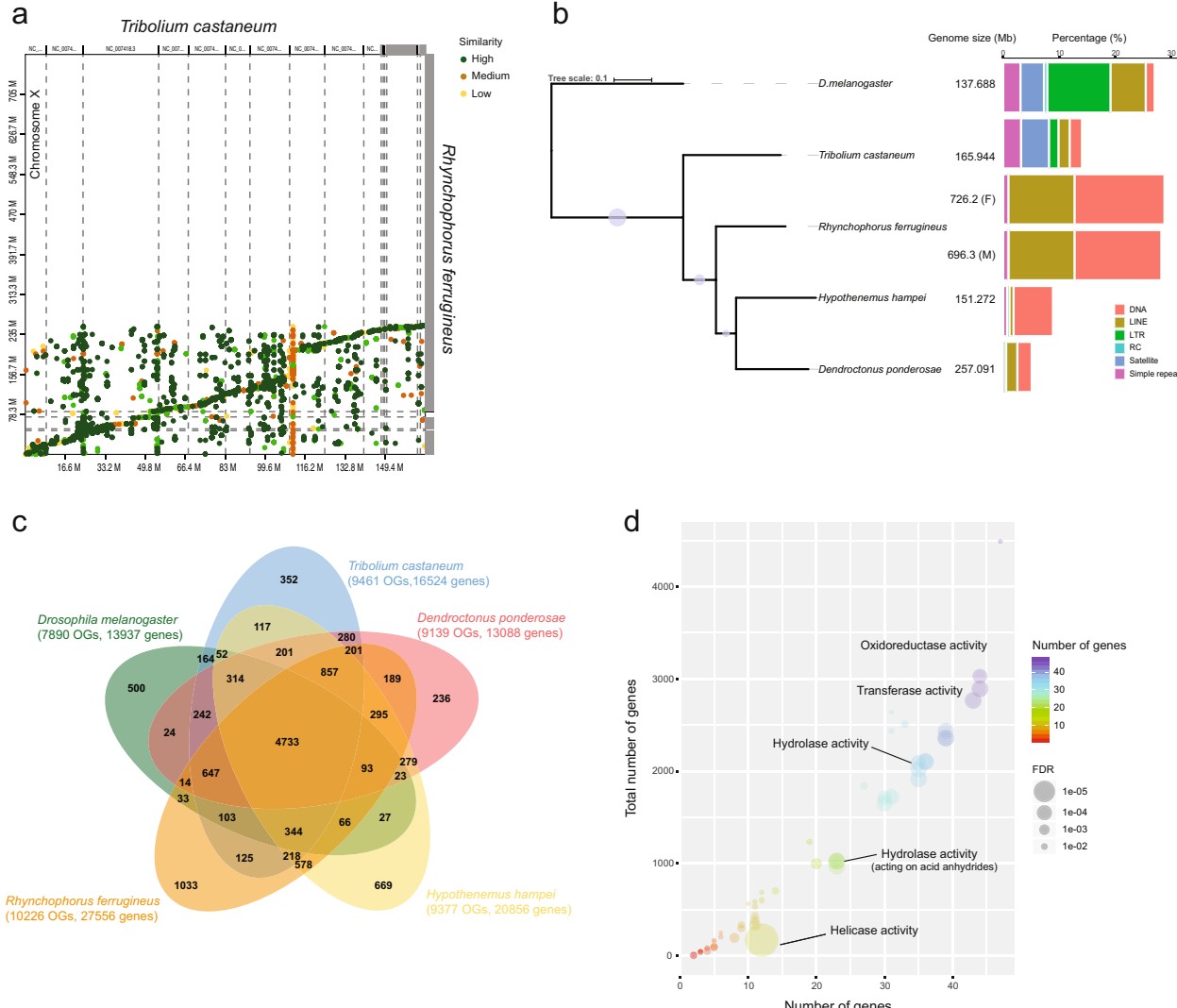

**Fig. 2 Comparative genomics and analysis of orthology. a** Synteny plot between the red palm weevil *R. ferrugineus* and the red flour beetle *T. castaneum* showing synteny going from low (yellow), medium (brown) to strong (green). **b** Vendiagram of shared and unique orthologues among *R. ferrugineus*, *T. castaneum*, *D. ponderosae*, *H. hampei*, and *D. melanogaster*, Orthologous groups (OGs) as well as the number of genes for each of the species is highlighted between brackets. **c** Enrichment of shared orthologues among the five species where the number of genes is shown as heatmap going from low (red) to high (purple) and significance with false discovery rate (FDR) depicted by the size of the bubble, small (low) to high (big). **d** The percentage of each transposable element is shown in each of the five species as a horizontal stacked barplot, with the genome size also being shown. The color depicts the different types of transposable elements such as DNA, cut and paste (orange), long terminal repeat (LTR) (green) as well as other satellite and simple repeats.

**Transposable element content**. Inter-species variation of genome size is known to be the result of amplification, deletion, and rearrangements of repetitive DNA sequences[29]. As a result, the size of the genome is a function of repeat dynamics but also of the average size of introns and many other factors[30]. Comparative analysis of the landscape of transposable elements (TE) (Fig. 2b, Supplementary Fig. 8, Supplementary data file 2) of *R. ferrugineus* showed that its genome has more repeats (45.24%; 354,353,149 bp) compared to *D. ponderosae* (16.41 %; 41,501,291 bp), and *H. hampei* (16.79 %; 25,391,186 bp). These percentages of TE content vary in Diptera from 6% in the Antarctic midge (*Belgica antarctica*) to 58 % in *Anopheles gambiae*, while the Hymenoptera honeybee (*Apis mellifera)* and turnip sawfly (*Athalia rosae*) have less than 6%. The Orthoptera migratory locust (*Locusta migratoria*) has 58% of its genome occupied by TEs. The most abundant transposable element in *R. ferrugineus* is transposase-mediated cleavage (Tc) mariner (class II "cut and paste" DNA transposon) (Fig. 2b).

**Ontology analysis of the *R. ferrugineus* proteome**. Orthologous analysis using the proteins predicted in *R. ferrugineus* against those of *H hampei*, *D. ponderosae*, *T. castaneum*, and *D. melanogaster* showed 4733 orthologous groups that are shared among all these species (Fig. 2c). Molecular function ontology of predicted proteins in the shared clustered group shows enrichments of genes encoding hydrolase, metal-ion binding, oxidoreductase, peptidase, and transferase, which is also found in other insect-plant systems[31,32] (Fig. 2c, Supplementary data file 9). Genes annotated as metal-ion binding and oxidoreductase activity include several alcohol dehydrogenases as well as many cytochromes and CYPs, while genes with transferase activity (e.g., methyl group, glycosyl group, acyl group, phosphorus-containing, or other groups) included glutathione transferases and UGTs. Hydrolase activity genes including peptidases, serine proteases, serine/threonine phosphatases were also annotated. Structural constituents of chitin-based cuticle are also enriched and include

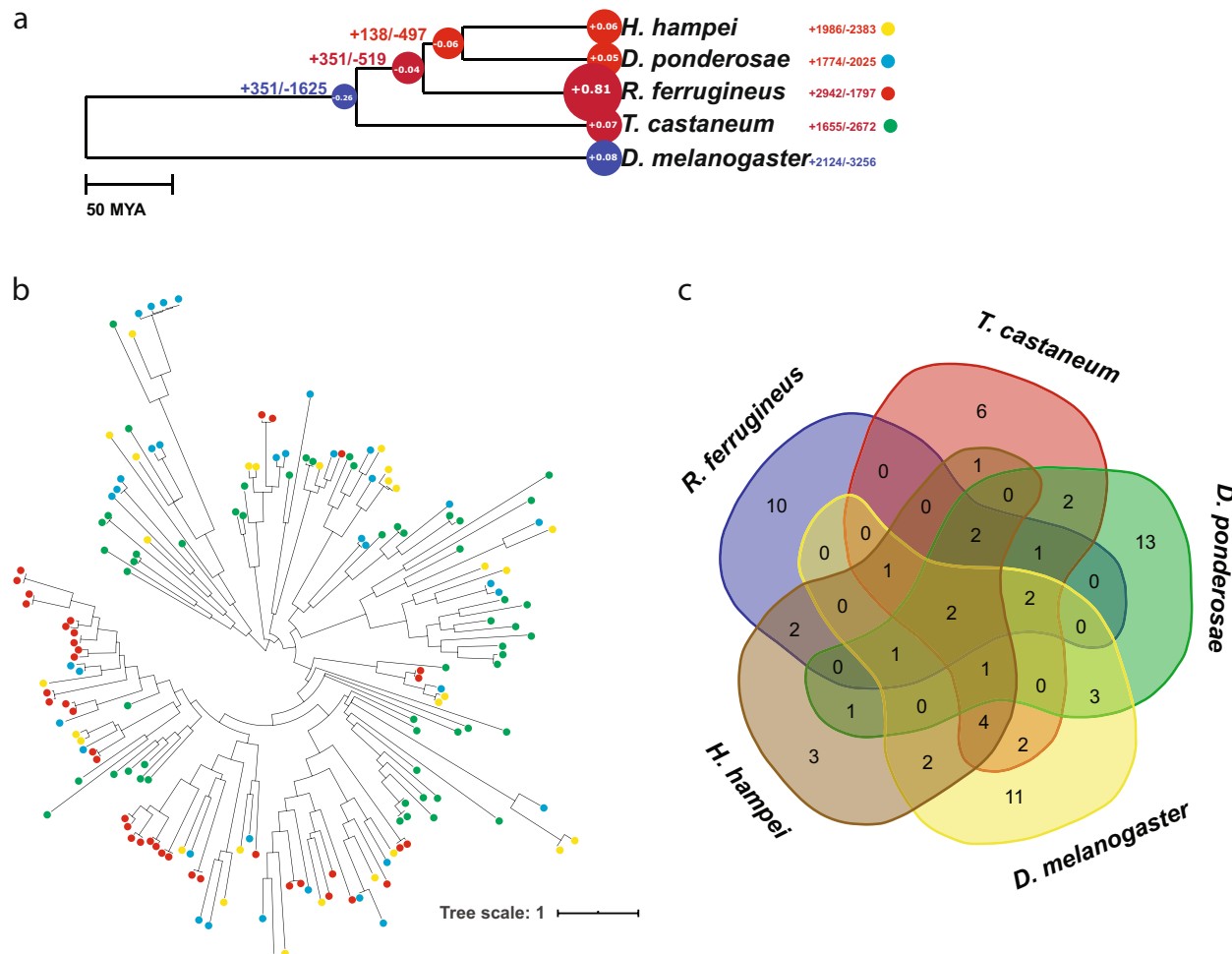

**Fig. 3 Evolutionary inferences of gene family sizes. a** We used CAFE (Computational Analysis of gene Family Evolution) to infer the size change of gene families. This summary tree shows the average expansion/contraction (radius of node circles), the number of expanded contracted families (±), and the estimated gene gain/loss rates (blue: low rate; red: high rate). **b** Phylogeny of Pheromone: Odorant biding protein (PBP:OBP) using *H. hampei* (yellow), *T. castaneum* (green), *D. ponderosae* (blue), and *R. ferrugineus* (red) showing the expansion and the diversity of this gene family. Colors correspond to colored dots in (**a**). **c** Vendiagram depicting the shared and unique families of PBP:OBP among the five species in the study.

insect cuticular proteins. Biological processes of this shared ontology were mostly enriched for signal transduction processes and lipid metabolism (Supplementary data file 9, Fig. 2d).

The Venn diagram in Fig. 2b shows that there are 1033 cluster groups; representing 2931 *R. ferrugineus* proteins that do not share direct orthology with other proteins (orphans). However 1554 (564 cluster groups) of these proteins still have recognizable functional domains. GO enrichment analysis of these clusters identified stress-activated protein kinase signaling cascades (JNK signaling pathway involved in insect immunity), replication fork processing and DNA recombination (including transposase activity) (Supplementary Table 1).

Transcription factors (TFs) play a vital role in controlling gene regulation and many diverse physiological processes in insects. Our comparative orthologous results show that *R. ferrugineus* has more bHLH, homeobox, Zf-C2H2 main families similar to other beetles in the study (Supplementary Fig. 9). Signaling and metabolic orthologous counts of protein involved in different pathway are summarized in Supplementary data file 10, which highlights some increase of protein in *R. ferrugineus* involved in the degradation of aromatic compounds, immunity, and metabolism of xenobiotics by cytochrome P450 pathways.

**Evolution of genes families in *R. ferrugineus*.** By clustering all the proteins into orthologous groups, we managed to identify *R. ferrugineus*-specific clusters and signatures of expanded and contracted gene families. For the expanded and contracted gene families, the CAFÉ results (Fig. 3a) are represented by the ultrametric tree where the average expansion/contraction is depicted by the radius of node circles. This tree highlights more expansion in *R. ferrugineus* compared to the other species in the Curculionidae family. There are 2942 expanded and 1797 contracted gene families in *R. ferrugineus* (https://github.com/LKremer/CAFE_fig) [–count_all_expansions] (Supplementary data files 11 and 12). There are 38 significant (*P* < 0.001) expanded and 4 contracted gene families (Table 2; Supplementary data file 13; zipped families). Below are examples of expanded/contracted families in *R. ferrugineus* in comparison with other members of the Curculionidae family (*H. hampei* and *D. ponderosae*).

In insects, perception of the environmental cues is mainly guided by chemical signals. The red palm weevil is an invasive species and similar to other insects, it relies mostly on its olfactory system for food foraging and for mating. It uses resistance mechanisms for detoxification of plant secondary metabolites and

**Table 2 Summary of gene families (expanded/contracted) in *R. ferrugineus*.**

| Gene families | Pfam | Function |
| --- | --- | --- |
| **Variant SH3 domain** | **PF07653, PF14604** | **Signal transduction related to cytoskeletal organization** |
| **PDZ domain** | **PF00595, PF17820** | **Signaling complex including neuronal synapses** |
| **Guanylate kinase** | **PF00625** | **Cell proliferation** |
| **GPCR proteolysis site** | **PF01825, PF16489** | **Mediate cell adhesion** |
| **Secretin family** | **PF00002** | **Immune system** |
| **Galactose binding lectin domain** | **PF02140** | **Immune system** |
| **RNase H-like domain** | **PF17919** | **Immune system** |
| **Integrase core domain** | **PF00665** | **Required for integration of viral DNA into host** |
| **Helix-loop-helix DNA-binding domain** | **PF00010** | **Developmental processes** |
| **Hairy orange** | **PF07527** | **Cell differentiation, embryonic patterning and other biological processes** |
| **SAM domain** | **PF07647, PF00536** | **Repressors of target gene expression and RTK signaling** |
| **HMG box** | **PF00505** | **Immune system** |
| **BAH domain** | **PF01426** | **Chromatin biology** |
| **Glutathione S-transferase** | **PF00043, PF13417** | **Development of insecticide resistance** |
| **N-terminal of Par3** | **PF12053** | **Cell polarity** |
| **Ion transport protein** | **PF00520** | **Signaling in all sensory modalities** |
| **Cyclic nucleotide-binding domain** | **PF00027** | **Cellular processes** |
| **PBP/GOBP family** | **PF01395** | **Odorant detection** |
| **RPEL repeat** | **PF02755** | **Actin binding** |
| **Sorbin homologous domain** | **PF02208** | **Signal transduction** |
| **Polo kinase** | **PF12474** | **Required for cytokinesis** |
| **RhoGEF domain** | **PF00621** | **Adaptation of cells to environmental signals** |
| **Cadherin domain** | **PF00028** | **Play a role in morphogenesis** |
| **Odorant receptor** | **PF02949** | **Odorant perception** |
| **Cytochrome P450** | **PF00067** | **Detoxification of natural and external chemical** |
| **Phorbol esters domain** | **PF00130** | **Pheromone response and communication** |
| **BTB/POZ domain** | **PF16017** | **Leg and antenna segmentation, sex differentiation, color sexual dimorphism** |
| **Autophagy-related protein** | **PF10377** | **Metamorphosis** |
| GNS1/SUR4 family | PF01151 | Glucose-signaling pathway |
| OAR domain | PF03826 | Correct morphogenesis of the limbs and cranium |
| Homeobox domain | PF00046 | Control of development and cell fate |
| Zinc carboxypeptidase | PF00246 | Protein digestion |
| Carboxypeptidase activation peptide | PF02244 | Protein digestion |

Bold color: expanded families.
Regular color: contracted families.

xenobiotic[33]. We highlight some gene families that are expanded in this context.

*Odorant receptor (ORs).* Olfactory gene families are involved in pheromone and odorant detection. They allow *R. ferrugineus* to locate infected palm trees that emit volatiles in the air, as well as the male aggregate pheromone released to coordinate explosive attacks[33]. The *R. ferrugineus* genome contains 46 OBP:PBP odorant/pheromone binding proteins (PF01395) and 80 ORs (PF02949). By comparison, *D. ponderosae* has 40 OBP:PBP and 57 ORs and *H. hampei* has 36 OBP:PBP and 16 ORs. In the same context, the generalist honeybees (*Apis mellifera* and *A. cerana*) have 21 OBP:PBP and 175 ORs for broad olfactory perception of pheromones blends and floral odorants. Similarly, *T. castaneum*, which is a pest for a broad range of dried stored products, has 56 OBP:PBP and 265 ORs[34].

A phylogenetic tree generated from the alignment of the different PBP:PBP shows their divergence, diversity, and expansion (Fig. 3b). There are two subclasses that are shared among all the species (Fig. 3c). There are ten unique OBP:PBP subclasses in *R. ferrugineus* (Fig. 3c; see also Supplementary data file 14). Two of the unique subclasses PBP:OBP in *R. ferrugineus* are important for host plant discrimination and to sense nutrient sources (with subclass 11)[35], while subclass 4 appears to be involved in male-specific pheromone production[36]. The similar numbers of OBPs in *R. ferrugineus* and in *D. ponderosae*, which are involved in the transport of odorants to ORs, is in contrast with the significant higher number of ORs in *R. ferrugineus*.

*Basic Helix-loop-helix (bHLH) DNA-binding proteins.* bHLH transcription factors play important roles in different developmental processes[37,38]. We identified an expansion of the Myc-type bHLH genes with a Pfam domain (PF00010) that are involved in cell proliferation/differentiation, sterol metabolism, adipocyte formation, and expression of glucose-responsive genes[39,40]. In this class of bHLHs, *R. ferrugineus* has seven members that contain a Sterol-sensing domain of the SREBP family, compared to *D. ponderosae* and *H. hampei* that only have four. In *Drosophila*, and mice, glucose can activate genes via the transcription factor ChREBP, an ortholog of the seven SREBP bHLHs found in the weevil (PF00010), to induce the utilization of glucose and de novo lipogenesis[41].

*Glutathione S-transferase.* GST is a large gene family that is involved in the detoxification of plant secondary metabolites. In *R. ferrugineus*, we identified 47 cytosolic GST genes, which is more than the 40 GST in *H. hampei* and 43 in *D. ponderosae*, the 38 GST found in *D. melanogaster* or 36 GST in *T. castaneum*. The higher number of GST in *R. ferrugineus* might reflect their need to detoxify diverse toxins, because their host range includes 26 species of palms. There are 6 subclasses of GSTs in *R. ferrugineus* with 4 members of the *Delta* family, 25 *Epsilon*, 8 *Omega*, 3 *Theta*, 4 *Sigma* and 1 *Zeta*. There is a correlation between the amplification of GST genes and resistance to insecticides[42,43].

We analyzed duplications/deletions, inversions and tandem duplications in *R. ferrugineus* (Supplementary data files 15 and 16). Our results show more duplication, in particular more

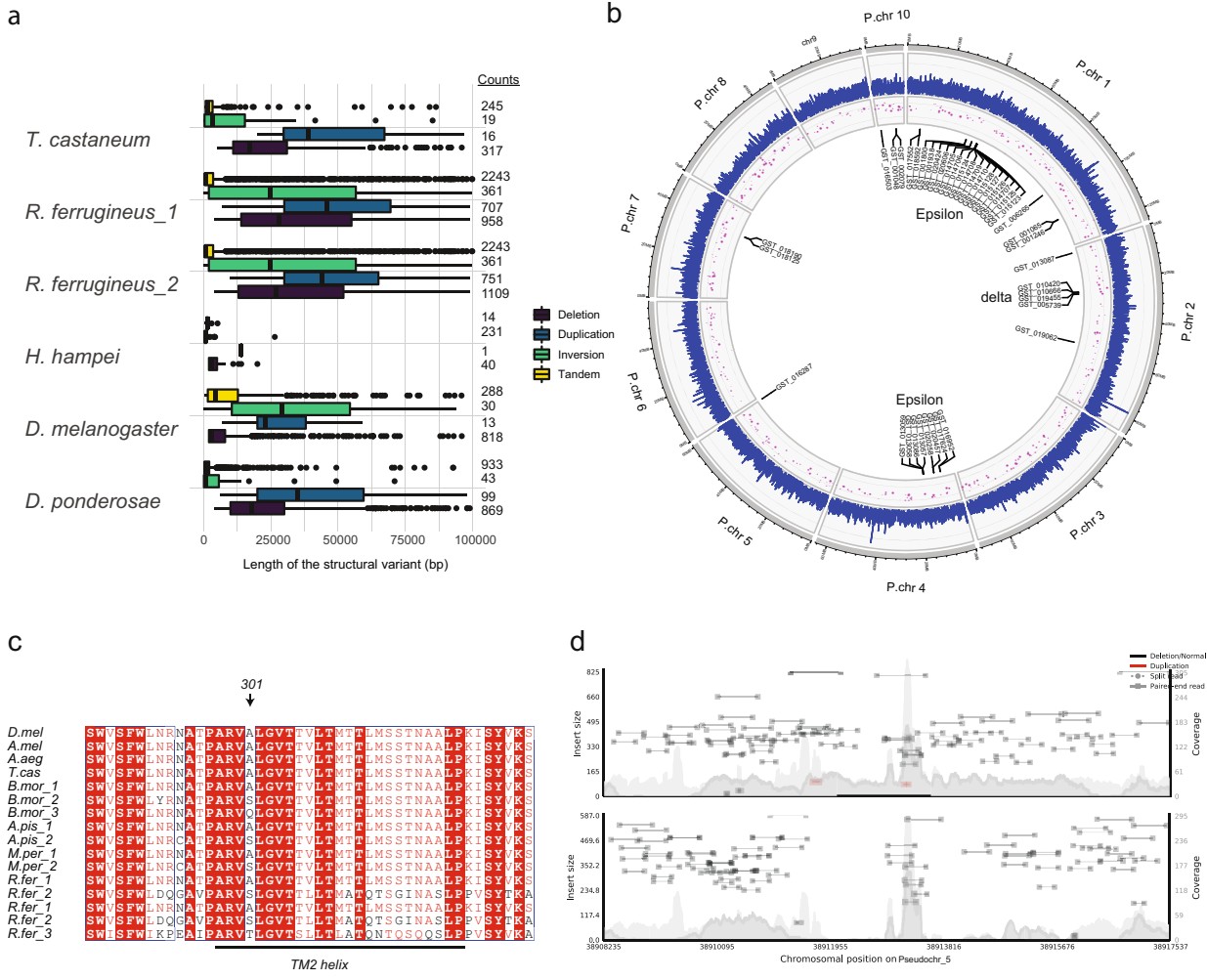

**Fig. 4 Structural variation. a** Distribution of the different classes of structural variants (SVs) (duplication, deletion, inversion, and tandem duplication) in the five species studied depicted in the horizontal boxplot with error bars represents standard deviation (s.d). The *x* axis represents the length of the structural variants with less than 100 kb. The counts of the SVs are represented in a column on the right side of the plot. **b** Circos plot depicting nucleotide diversity distribution across the different pseudochromosomes from the outer track of the plot. From the inside duplications, as the second track represented as the normalized read depth, in red bubble. The size of the bubble represents the size of the duplication in kilobases (kb). The inner part of the Circos is the annotation of the tandem duplication of glutathione S-transferase (GST) highlighting the cluster of Epsilon and delta subclasses. **c** Alignment of the portion of Dieldrin gene Rdl surrounding the equivalent residue 301 in insect species, showing copy number variation and the relative amino acid variation at the 301 site. *Bombyx mori* contains three Rdl orthologs, with a different residue at 301 (1 = Ala, 2 = Ser, 3 = Gln). Both aphid species contain two Rdl orthologs: Acyrthosiphon pisum and Myzus persicae (1 = Ala, 2 = Ser). *Rhynchophorus ferrugineus* (1 = Ala, 2 = Ser, 3 = Thr). D. mel, *D. melanogaster*; A. mel, *Apis mellifera*; A. aeg *Aedes aegypti*, T. cas *Tribolium castaneum*, B.mor *Bombyx mori*, A. pis *Acyrthosiphon pisum*, M. per *Myzus persicae*, *Rhynchophorus ferrugineus* R.fer_(1,2,3). **d** Depth of the coverage plot for two individuals showing the duplication (in red) of the Rdl gene.

tandems duplications (2243) relative to the other beetles in our study (Fig. 4a) (Fisher's exact test *P* < 0.001). The intersected results of the structural variants (SV) with the *R. ferrugineus* genome annotation revealed several genes families important in detoxification of secondary metabolites and insecticides that were tandemly duplicated such as GST and P450 (Fig. 4b, Supplementary Fig. 10). Thirty-eight of the 45 *R. ferrugineus* GST mapped to 4 of the 10 pseudochromosomes (Fig. 4b). Tandem duplication is a general feature for GSTs, including in the beetles. In *R. ferrugineus*, the GSTs are divided into clusters, according to their subclasses. Four Delta are in a cluster on pseudochromosome 2 and 25 Epsilon are in two clusters on pseudochromosome 1 and 4 (Fig. 4b). In *D. melanogaster*, the ten Epsilon subclasses as well as the Delta genes are all in tandem[44]. In *T. castaneum*, the two members of the Delta subclass are in tandem and genes in the Epsilon class are in tandem in two clusters on chromosome 2 and 3[45].

**Cytochrome P450.** The Cytochrome gene family encompasses oxidases and cytochrome P450 (CYP450) monooxygenases. It has a wide diversity of functions for steroid hormone synthesis, which is important for the development and reproduction of insects, to the metabolism of chemicals that play a role in host plant adaptation and survival in toxic environments. A number of related P450 proteins control these processes with the number and rate of expansion of Cytochromes dependent on the species physiology and the environmental in which a species lives. In *R. ferrugineus*, the cytochrome P450 family is composed of 120 genes compared to 88 in *D. melanogaster* (http://flybase.org/). CYP6 and CYP12 (mitochondrial) are the most expanded gene families and *R. ferrugineus* has 104 CYP6 genes (and 16 CYP12), almost 5 times the 23 genes found in *D. melanogaster*, with a more specific expansion of subfamilies CYP6A (14 genes), CYP6G (9 genes) and CYP6D (5 genes), which have been associated with insecticide resistance[46,47]. The number of these genes

**Table 3 Parameter estimates and likelihood scores for glycoside hydrolase (GH16) gene under models of variable $\omega$ ratios.**

| Nested model pairs | $d_N/d_S$[b] | Parameter estimates[c] | PSS (*$P > 95\%$; **$P > 99\%$)[d] | Likelihood |
|---|---|---|---|---|
| M0:one-ratio (1)[a] | 0.0871 | $\omega = 0.0871$ | | −12,014.227 |
| M3: discrete (5) | 0.2194 | $p_0 = 0.335$, $p_1 = 0.353$, ($p_2 = 0.298$) | 3 I 0.999** | −11,619.479 |
| | | $\omega_0 = 0.011$, $\omega_1 = 0.082$, $\omega_2 = 0.251$ | 6 W 0.979* | |
| | | | 9 I 0.999** | |
| M1: neutral (1) | 0.3076 | $p_0 = 0.764$, $p_1 = 0.235$ | | −11,844.804 |
| | | $\omega_0 = 0.094$, $\omega_1 = 1$ | | |
| M2: selection (3) | 0.3076 | $p_0 = 0.764$, $p_1 = 0.085$, ($p_2 = 0.149$) | 3 I 0.956* | −11,844.804 |
| | | $\omega_0 = 0.094$, ($\omega_1 = 1$), $\omega_2 = 1$ | 6 W 0.730 | |
| | | | 9 I 0.945 | |
| M7: beta (2) | 0.1221 | $p = 0.605$, $q = 3.970$ | | −11,628.839 |
| M8: beta $+ \omega > 1(4)$ | 0.2225 | $p_0 = 0.989$, ($p_1 = 0.010$) | 3 I 0.999** | −11,617.599 |
| | | $p = 0.628$, $q = 4.589$, $\omega = 10.628$ | 6 W 0.972* | |
| | | | 9 I 0.998** | |
| M8a: beta $+ \omega = 1(4)$ | 0.1281 | $p_0 = 0.652$, ($p_1 = 0.028$) | | −11,621.768 |
| | | $p = 0.652$, $q = 5.181$, $\omega = 1$ | | |

| Gene | Model[e] | P value |
|---|---|---|
| GH16 | M3 vs M0 | 2.86508957212e−167 |
| GH16 | M2 vs M1 | 1 |
| GH16 | M8 vs M7 | 1.31253379663e−05 |
| GH16 | M8 vs M8a | 0.00387888988444 |

[a]The number of free parameters in the $\omega$ distribution.
[b]Average ratio dN/dS of all sites for the GH16 gene alignment.
[c]The number in parentheses are not free parameters.
[d]Number of positively selected sites.
[e]Likelihood ratio test statistics for models of variable selective pressure among codons.

is even higher in *H. hampei* (116 CYP6 and 12 CYP12) and in *D. ponderosae* (117 CYP6 and 8 CYP12) showing that a dramatic expansion occurred in the Curculionidae in which it could mediate resistance to insecticide[33]. All pseudochromosomes, except #10, had genes for P450s. Only 8 P450 genes were found individually located, while the other 112 are tandem duplicates (Supplementary Fig. 10).

Other gene families, including several immunity gene families (e.g., secretin family) (Supplementary data file 11), were also expanded, suggesting that the *R. ferrugineus* immune system responds differently to the challenge of the life inside a tree.

*Glycosyl hydrolase and carboxypeptidase genes.* The genome-wide comparison of digestion-related genes such as proteases suggests that they have undergone a major expansion in Diptera, Lepidoptera, and Coleoptera, but not in Hymenoptera or Hemiptera[48]. In contrast to *T. castaneum* that has 62 carboxypeptidase genes, *R. ferrugineus* has only 13. There are 20 in *D. ponderosae* and 30 in *H. hampei*. However, the genome of *R. ferrugineus* encodes 70 glycosyl hydrolase genes, which might be important for the hydrolysis of the rich sugar content in the phloem sap that is rich in sugar but lacks starch. *T. castaneum* that feeds on starch-rich grains has a high number of α-amylase genes (12) compared to *D. ponderosae that* has 8 genes and *H. hampei* that has 6 genes. In contrast, we found 17 α-amylase genes and one chitin synthase CHS2 gene in *R. ferrugineus*.

**Gene families under positive selection.** We conducted phylogenetic tests for selection in *R. ferrugineus* by aligning CDS sequences from each gene family to their homologs in *D. ponderosae* and *H. hampei*. We found evidence of positive selection in 115 gene families (Supplementary data file 17). These functions of these families range from calcium channels (e.g., TRPM) to xenobiotic metabolism (e.g., CYP450, UDP-glucuronosyltransferase), to odorant-binding proteins. One interesting gene family that is under positive selection is family 16 of glycoside hydrolase (GH16)[49] that supports selection for this family ($P = 0.003$) (Table 3). Another gene family showing signature of positive selection is the TRPM transient receptor potential ion channels (Supplementary Table 2), and the GABA-gated chloride channel subunit encoded by the Rdl gene (Fig. 4c, Supplementary data file 17) that have been shown to be responsible for insecticide resistance in many insect species[50,51]. In *R. ferrugineus*, we identified a 10 kb duplicated region containing a WT copy of Rdl and a second copy with the A30S point mutations as well as another A301T (Fig. 4d). The frequency of the mutation using 50 individual transcriptomes is high (60%) (Supplementary Table 3).

**Ancestral and present effective population size (Ne).** We applied PSMC analysis (https://github.com/lh3/psmc) to evaluate population dynamics of *R. ferrugineus* from 8 million years before present (Ma) to 10,000 years before present (ka). Assuming a generation time of ~4 months, we estimated a per nucleotide per generation mutation rate of $0.89e^{-10}$ (Supplementary data file 18 (zipped files for each pseudochromosome)). We estimated the peak of Ne at $5.5 \times 10^{+06}$, which occurred approximately around 1.2 million years before present (Ma). This expansion was followed by a population decline from the middle to the end of Pleistocene (Fig. 5).

In order to test for the present effective population size, we used the gene for Rdl, which is known to have two single nucleotide mutations in the same codon that confer resistance to insecticides in other insects[50] (A→S) (Fig. 4c). The frequency of the mutation (A→S) in the population was determined (Supplementary Table 3) to be ~0.6. Assuming both bp mutations are required for the resistance, and a base pair mutation rate similar to the red flour beetle of $\approx 2.70e^{-10}$, this gives a mutation rate of $\approx 5.39e^{-10}$. If we apply the haploid algorithm to the data[52], we can estimate diversity $\theta = 0.005$ (giving $n_s = 100$ (2 chromosomes and 50 individuals), $n_m = 68$) and a recent effective population size of $2.3 \times 10^{+06}$.

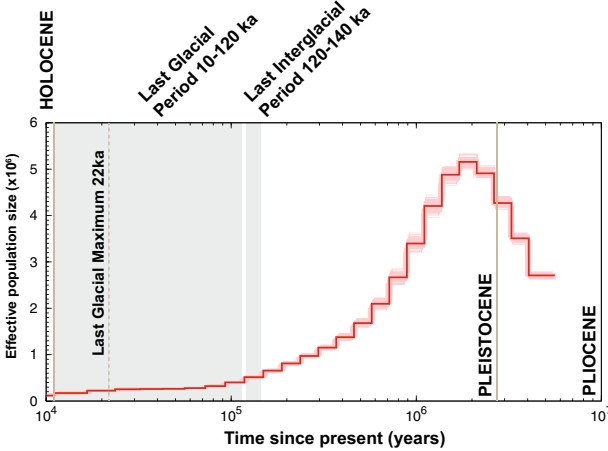

**Fig. 5 Demographic history of R. ferrugineus.** PSMC analysis was applied on the genomic sequences of *R. ferrugineus* converted to demographic units assuming a generation time of 4 month ($g = 0.3$ years) and a substitution rate of $\mu = 0.89 \times 10^{-10}$. The *x* axis represents time before present in years in log scale and the *y* axis is the effective population size. The bold red curve shows the estimate of the original data and the shaded red curves are estimate for 100 bootstrapped sequences.

**Horizontal gene transfer (HGT).** Genes involved in plant cell wall degradation found in some insects were often horizontally transferred from bacteria[53]. CAZy (Carbohydrate active enzymes (http://www.cazy.org)) are the enzymes that collectively assemble and degrade oligo- and polysaccharides. We identified 224 GHs assigned to 19 families in *R. ferrugineus* (Supplementary Table 4).

Models of horizontal transfer suggest that successful acquisition of a horizontally transferred gene requires that the gene be active and maintained, and likely be under positive selection[54]. GH16 in *R. ferrugineus* is under positive selection and responsible for the hydrolysis of β-1,3 glycans, which are found in the phloem of plants as callose[55] and are a main component of the date palm sap[56].

GH16 could thus have originated from HGT from bacteria or fungi, although endogenous eukaryotic β-1,3-glucanases such as Gram-negative binding proteins (GNBPs/βGRP) already exist in insects[57]. We investigated whether GH16 was also horizontally transferred into the *R. ferrugineus* genome. Phylogenetic analysis shows that GH16 from *R. ferrugineus* clusters monophyletically with *D. ponderosae* but is distantly related to *T. castaneum* GH16 (Fig. 6a, Supplementary Fig. 11). No match was found in *H. hampei*. The clustering illustrates also the duplication of the GH16 in *R. ferrugineus*. The phylogeny also shows clear separation of the eukaryotic GNBPs[57] (Fig. 6a) found in many order. In addition, this cluster was closely related to a gene both in a γ-proteobacterium as well as in the fungus *Pisolitus microcarpus*, which did not allow us to identify its origin from a fungus or a bacterium (Fig. 6a). Using RNAseq, we validated the expression of these GH16 genes (Fig. 6b). We also used a fragment from one of the GH16 genes and used PCR to validate its presence in the genome in different parts of the weevil to rule out contamination coming from the gut microbiota of *R. ferrugineus* (Fig. 6c, Supplementary Fig. 12).

GH16 could have been acquired from a bacterium or from a fungus since the closest fungal relative showed no introns. If the gene was acquired once in the last common ancestor, GH16 should share the same exon/intron structure in *D. ponderosae*, in *R. ferrugineus*, and in *T. castaneum*. However, GH16 has no introns in *D. ponderosae* and has three in *T.*

*castaneum*. In *R. ferrugineus*, the number of introns varies from 2 to 7, suggesting independent and subsequent acquisition of introns after HGT. We evaluated the expression of the different GH16 in *R. ferrugineus*, which showed higher gene expression for the genes with more introns (Supplementary Table 5). This suggests that the introns were acquired after HGT followed by gene duplication.

## Discussion

The genome of *R. ferrugineus* provides insights into the behavior of the species. It is the largest beetle genome of the Curculionidae family sequenced to date with an estimated genome of around 720 Mb. It shows a high synteny with *T. castaneum* although the two species diverged 236 Mya ago[58] and has low diversity on the X chromosome, likely due to the suppression of recombination as shown in other insects[59].

The increased number of genes in *R. ferrugineus* genome compared to other beetles results from the expansion of gene families. In contrast, the average intron length in *R. ferrugineus* is lower when compared to other Curculionoidea, which is not consistent with the established correlation between genome size and intron length[60] (Supplementary Fig. 6). This suggests relaxed selection and an increase in deleterious TEs that represent 45% of the weevil genome, leading to structural variations such as duplications, deletions, inversions, and translocations[61–63].

Orthologous analysis shows enrichment of genes with transferase and hydrolase activity that are important for detoxification, xenobiotic metabolism, and digestion, while feeding on different host plants[17,64,65]. The genome shows expansion of gene families important for chemoreception, food intake and for dealing with a hostile environment (ORs and bHLHs transcription factor (TF)). This suggests that the large number of host plants for this generalist species leads to diversification of ORs. The function of the expanded family of bHLHs transcription factor is still unknown, but members of this family regulate glucose metabolism and the production of pheromones[66,67] involved in the massive invasion of trees and promote mating. TFs are essential in orchestrating many physiological processes and their identification will help the growing entomologist, to invest more in this understudied filed by using emerging research methods to study their regulatory functions. Furthermore, it will prompt to investigate more their contributions in pest control and in general human health.

The vast array of GST and CYP450 genes in insects represents the largest repertoire of detoxification genes known. In particular, the Epsilon and Delta subclasses of GSTs are involved in insect response to environmental conditions[14], as well as in xenobiotic and insecticide resistance[14]. For instance, *A. gambiae* and *A. aegypti* are able to metabolize DDT by GST epsilon2–2[68,69]. We propose that the history of insecticide application for the control of *R. ferrugineus*[70] has led to resistance mediated by amplification of GST[33]. It is known that the P450 families evolve through duplication and diversification[71,72]. Our structural variation results suggest that P450 in *R. ferrugineus* also arose through localized tandem duplication[73,74]. *R. ferrugineus* has undergone different insecticides treatments[70], and this explains the rapid evolution and duplication of GST and P450 monooxygenases as well as the TRPM family that could promote insecticide resistance (Supplementary data file 18). The recent duplication of the gene that encodes the Rdl channel that is the target of cyclodiene and phenylpyrazole insecticides, likely results from their use against the native and invasive *R. ferrugineus*[75]. Insecticide resistance can be accomplished via gene duplication by increasing the Rdl gene product or to adaptive mutations in Rdl that prevent the action of the insecticides without affecting its essential role[50]. The resistance to cyclodiene dieldrin in *Drosophila* is due to a single amino

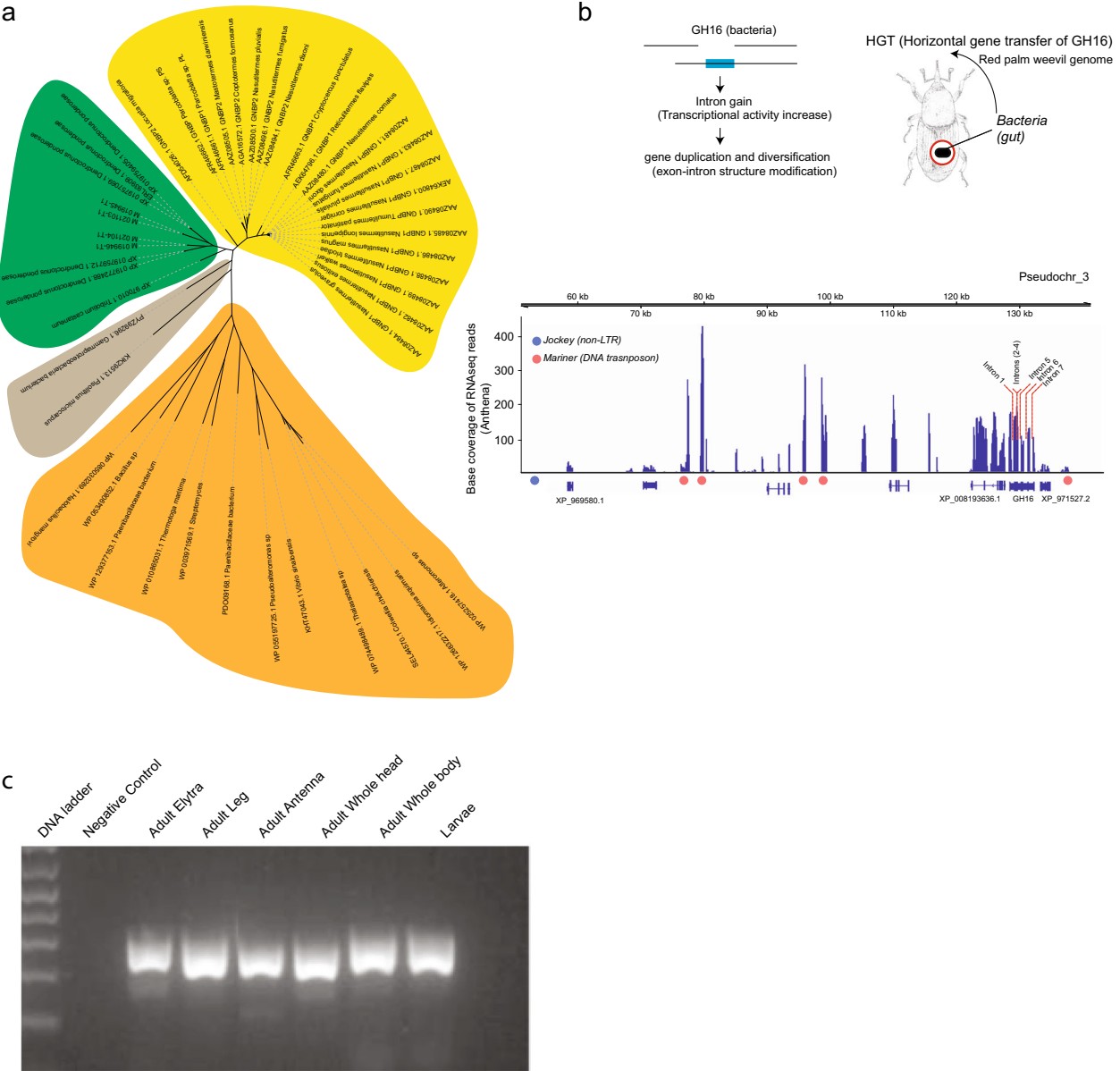

**Fig. 6 Horizontal gene transfers in *R. ferrugineus* (HGT). a** Phylogeny of the some of the different glycosyl hydrolase (GH16) (e.g., M019946-T1) that are horizontally transferred and clustered with other beetles, which are highlighted in green. The Yellow cluster highlight the Eukaryotic Gram-negative-biding protein, similar to GH16, but already exists in insects. The gray cluster highlights the close microorganisms (bacteria and fungi) to the horizontally transferred hydrolase. While the orange cluster shows the distantly cluster of microorganisms. **b** The schematics about HGT depicted by acquisition, intron gain, and duplication and diversification. We show a GH16 with seven introns gains as well as expression on the y axis shown as coverage of RNAseq reads, where the surrounding shows DNA and non-LTR transposable elements. **c** Gel electrophoresis of a 258 bp fragment of one GH16 to validate the presence in the genome and rule out gut microbial contamination.

acid replacement, A30S[50], a mutation that was subsequently identified in Rdl orthologs in different resistant insect species (Fig. 4c). This classical example of parallel evolution of the Rdl gene shows that it is a hotspot of evolution (Fig. 4c). Our data show that positive selection is currently acting on a mutation in Rdl in *R. ferrugineus* that is found at high frequency through the population, but it is not clear whether this mutation(s) promote resistance. This mutation might explain the inefficacy of cyclo-diene dieldrin insecticides to control *R. ferrugineus*.

In *R. ferrugineus*, the contraction of carboxypeptidase genes may reflect the feeding strategy of its larvae that spend most of their lives inside the trunk of the date palm, chewing and sucking the sugar-rich sap of the soft tissue. This lifestyle minimizes the requirements for digestion of proteins. The 17 α-amylase genes

we found likely help the larva to ingest the starch-rich date palm stem, which is the point of entry for adults during invasion. *R. ferrugineus* has only one chitin synthase CHS2 gene that produces the peritrophic matrix, a chitin layer that lines the midgut and protects the epithelium from damage caused by rough food particles, digestive enzymes as well as ingestion of pathogens. In contrast to sucking species like *R. ferrugineus*[48], these genes are expanded in species feeding on diverse grains (*T. castaneum* has 3 genes) or woody substrates (*D. ponderosae* has 5 and *H. hampei* has 2).

The PSMC analysis suggests that *R. ferrugineus* experienced an increase in Ne during the early Pleistocene, which could reflect an expansion and spread of the range of the ancestral population. The duration of the last glacial period and the transition to

Holocene was, for most of the species, associated with dramatic population changes associated with reduced Ne. The mid to the end of Pleistocene experienced glacial-interglacial cycling and around 900 ka, there is an evidence that the two southern oceans experienced sluggish thermohaline overturn[76], which might have led to the contraction of the range of the Pacific and India coconut trees (Cocos nucifera) in isolated refugia. This scenario is consistent with the distribution of the two native species R. ferrugineus and R. vulneratus[4] as the insect-tree relationship must have resulted in the contraction of the weevil population.

Population genetic analysis could enable the establishment of the possible route(s) of invasion from South East Asia to the Mediterranean and to the Arabian Gulf. For example, the recent (1985)[77] R. ferrugineus introduction to the United Arab Emirates likely originated in a date palm offshoot from an infected country[78], which created a bottleneck effect for a period of time, followed by an increase in the effective population size that likely reduced drift and increased the effect of selection at fixing beneficial mutations[79]. The difference in the calculated effective population size and the actual number census population size (actual numbers of animals present) ($\sim 10^8$) in R. ferrugineus is likely caused by overlapping generations[80,81].

GH16 genes are under positive selection and provide a significant advantage to the weevil to efficiently process its food: Since the larvae are embedded inside the truck of the date palm tree, these genes are likely very important for larval digestion and survival. Our phylogenetic clustering with D. ponderosae and T. castaneum suggests that GH16 in R. ferrugineus originated from HGT that occurred in one of the common ancestors of beetles, although we cannot rule out its independent acquisition. After HGT, natural selection played a role in maintaining and fixing GH16 in the population of R. ferrugineus and later expanding it by gene duplication. Another example of GH specialization is in the western corn rootworm (Coleoptera Diabrotica virgifera virgifera) that has a close relationship to maize[82]. GH16 was shown to be horizontally transferred from bacteria in the Antarctic springtail, Cryptopygus antarcticus[83]. Other phytophagous beetles and some leaf beetles also acquired plant cell degrading enzymes via HGT, such as GH28, GH45[19,84].

This high-quality whole genome assembly provides a foundation that will make it feasible to genetically modify the insect in order to potentially control this pest by editing genes important for the reproduction of R. ferrugineus through the eventual release in the population of mutants successfully tested in field trials.

## Methods

**R. ferrugineus samples.** Male and female adult (Fig. 1a) were collected from an infested field in Al Ain, UAE region (Fig. 1c), flash frozen and maintained on dry ice for sample extraction, library preparation, and sequencing. High molecular weight DNA was extracted. Oxford Nanopore and 10X Genomics libraries were generated as well as two Hiseq 2500 (2× 150 bp) libraries following standard protocols before sequencing. These data generated 190 Gb of raw sequencing reads with ~80× coverage. RNA extraction was done on 50 individuals. Library construction was done using a TruSeq RNA Library Prep Kit v2 (Illumina, San Diego, CA, USA) and sequenced in a 150 bp PE run on an Illumina HiSeq 2500 platform. The transcriptome data were used for genome annotation and expression analysis. In addition, available short-reads archive (SRA) RNAseq libraries were used (SRX096967, SRX096966, SRX096965, SRX096462, SRX096968, SRX096969, SRX096970, SRX096971, and SRX096972) for different developmental stages (egg, larvae, and pupae) in the annotation and expression analysis.

**Genome size estimation.** We used two methods to estimate the size of the R. ferrugineus genome. For flow cytometry, we dissected whole brains and placed them in 1 mL Galbraith buffer[85] along with the head of one Drosophila virilis female. The mixture was grounded to release the nuclei and filtered through 40-μL nylon mesh being mixed with a vortex, and stained staining with 25 μL propidium iodide for 3.0 h at 4 °C. The relative fluorescence of 2 C nuclei from the sample and standard were measured with a Beckman/Counter CytoFLEX flow cytometer. The amount of DNA in each sample was determined as the ratio of the average relative fluorescence of the diploid sample nuclei divided by the relative fluorescence of the

diploid nuclei of the standard, divided by the amount of DNA in the D. virilis standard. Linearity of the CytoFLEX was verified by estimating comparable genome size using the 4 C peak of the sample and standard. Six biological replicates and three technical replicates were tested.

We also used a Kmer-based approach to estimate genome size and heterozygosity using Illumina Hiseq 2500 (2× 150 bp) paired-end sequencing data[86].

**Genome assembly and annotation.** Genome assembly was done using a combination of 10X genomics and Illumina Hiseq 2500. The results were combined with Oxford nanopore. The final assembly was done using syntheny to the T. castaneum (see Supplementary methods in Supplementary information). Genome annotation was carried out using Funannotate (https://github.com/nextgenusfs/funannotate), a gene prediction pipeline (see Supplementary methods in Supplementary information). We compared R. ferrugineus genome features to other beetles' annotation (D. ponderosae, H. hampei and T. castaneum) using gt stat command from genometools[87] and extracted information from their annotation file format GFF3(s) associated genomic features (genes, exons, CDS, intron) length and counts. We annotated TFs using (http://bioinfo.life.hust.edu.cn/AnimalTFDB/) and signaling and metabolic proteins using KEGG pathways reconstruction (https://www.genome.jp/kegg/tool/map_pathway.html).

**Genome characterization of repeats.** We used RepeatModeler a de novo prediction analysis (http://www.repeatmasker.org) to construct a transposable element library specific for the species. Repeat-Masker (4.0.7)[88] was used for TE identification and classification using the generated library by RepeatModeler. Repeats with percentage genome content for D. ponderosae, H. hampei, and male/ female R. ferrugineus was generated. Repeat Landscape of the different classes (LTR and non-LTR, DNA-TE) is plotted for male and female of R. ferrugineus using R[89].

**Phylogenetic analysis and gene family evolution.** Predicted proteins encoded by the four insects genomes with GeneBank assembly accessions (GCA_000001215.4; GCA_000002335.3; GCA_000355655.1; GCA_001012855.1) and the one from R. ferrugineus were filtered to keep the longest isoforms using a script provided in CAFÉ. Orthologs in the R. ferrugineus genome in comparisons to three beetles' genomes as well as D. melanogaster were generating using Orthofinder with default parameters. A phylogenetic tree using 439 single-copy ortholog groups was generated in orthofinder after supplying –m MSA (command line to generate alignment file). The number of conserved sites was calculated for the concatenated alignments.

Using the phylogenetic tree generated above, we have generated a calibrated species tree using the software r8s (http://loco.biosci.arizona.edu/r8s/) and the analysis was done using the penalized likelihood method and the TN algorithm. D. melanogaster and the mountain pine beetle (D. ponderosae) were chosen as calibration points using (http://www.timetree.org). We used CAFE[90] program version 3.1 for gene family expansion/contraction across the phylogeny as well as estimated the gene gain/loss rates varying lambda (maximum likelihood value of the birth and death parameter) value across the branches where each branch has assigned unique lambda and the best value was obtained using iterative calculation. Significant size variance of expansion and contractions of gene families was identified using 1000 random samples and a p value of 0.01 and deviated branches were identified using Viterbi algorithm implemented in CAFE with a p value of 0.05. Phylogenetic tree was build using an online tool (http://www.phylogeny.fr/simple_phylogeny.cgi) using protein alignment of an expanded family using matches from the different species analyzed.

**Structural variation.** Normalized Read-depth variation analysis was performed using CNVnator[91] (version 0.2.7). Aligned bams were used as input for CNVnator to extract read alignment information. A bin size of 1 kb was used in the intermediate processing of the bams as well as when calling variants. A table of duplication and deletion is generated. We discarded any duplication/deletion more than >100 kb as well, as hits that span gaps and beginning of a scaffold. Tandem duplication was screened using the software SoftV[92]. (See Supplementary methods in Supplementary information). We intersect duplication and tandem with the annotation for male and female and looked at genes that overlap the structural variant. Duplication/Tandem duplication was plotted using Circos[93].

**Horizontally transferred gene families under positive selection.** We investigated the extent of HGT using similar approach as in Nowel et al.[94] (see Supplementary methods in Supplementary information).

We looked if there were any selective pressures on different gene families by aligning sequences from R. ferrugineus, the mountain pine beetle and coffee borer. We ran the Fustr[95] with command Codeml from PAML[96]. A list of gene families under positive selection is reported. We highlighted two examples; one is the glycoside hydrolase (GH16) gene and the other is the transient receptor potential ion channels (TrpmM) and estimated parameter and likelihood scores for under models of variable $\omega$ (dN/dS) ratios. dS represents synonymous rate while dN non-synonymous rate. In the absence of evolutionary pressure this ratio $= 1$, under purifying selection it is $< 1$ and under positive selection it is $> 1$.

Phylogenetic analysis of GH16 was done using a combination of BLAST[97] and available CAZy databases (http://www.cazy.org/GH16_unclassified.html) in order to gain insight into its evolutionary history. We looked at the potential donor HGT microorganism, using the build up phylogenetic tree of the target with the most probable species that share maximum homology with the GH16 domain using blast to microbiome.

**Ancestral and recent effective population size (Ne).** The mutation rate for *R. ferrugineus* was estimated comparatively using the red flour beetle genome assembly (*T. castaneum*), downloaded from NCBI public database (https://www.ncbi.nlm.nih.gov/genome/?term=txid7070[orgn]), Tcas5.2[34] (Tribolium Genome Sequencing Consortium). The genome was split into individual chromosomes and the *R. ferrugineus* genome was aligned to each one of the red flour beetle chromosomes using LastZ v1.04.00[98], applied with the following parameters: −ydrop = 9400, −hspthresh = 4500, −gappedthresh = 3000 and −notransition. The number of *nmatch* (matches) and *nmismatch* (mismatches) was used in the output format options, respectively. Any matches or mismatches were not considered if they are classified as N/n or if there is an alignment gap. The mutation rate (per nucleotide per year, u) was calculated using the equation: (number of mismatches/total length)/2$t$, where $t$ is the divergence time between the red flour beetle and *R. ferrugineus*, which is estimated around 236 Mya[57].

We ran PSMC (https://github.com/lh3/psmc) analysis using consensus genome sequence (fastq) that was filtered for coverage and sequencing errors. For each *R. ferrugineus* sample, we used samtools to generate the consensus autosomal fastq using the "mpileup" command. SNP calling on single individual was done using samtools pipeline, which is independent of population frequencies and not assume Hardy–Weinberg equilibrium. We masked sites when read depth of a site is less than third the average depth genome. These criteria represent the default setting in PSMC when analyzing highly covered genome, which is our case.

We adjusted some parameters (−t, −p and −r) and set the upper limit of TMRCA to 5 with the −t option, −r option to 1 ($\theta/\rho$). The analysis of effective population was inferred using 24 free atomic time intervals (4 + 24 × 2 + 4 + 6 + 10) and this was set with −p option. We performed 100 bootstrap replicates to check for variance in effective population size (Ne). This was done on a 5 Mb sequences obtained from the consensus genome using the splifa command in PSMC. We applied the mutation rate calculated above. Diversity estimate $\pi$ (Pi) and $\theta$ (Watterson) was generated from the whole genome of the *R. ferrugineus* male and female using angsd[99] version 0.917 and plotted using Circos[93].

For recent effective population size (Ne), we followed the method in Khatri and Burt[52]. We calculated a recent Ne, where we used only the knowledge of the number of independent recurrent origins and the frequency of the beneficial allele in the population, without a prior knowledge of the strength of selection and age of mutation. In this analysis, we used the Rdl gene, which is known to have two point mutations in the same codon that confer resistance to insecticides, (A→S mutations) and the frequency is used from the data.

**Statistics and reproducibility.** All statistics was done using available packages and reproducibility can be accomplished using the same command lines mentioned in the methods, where we used for most of the analysis publicly available softwares and online tool for plotting with adjusted parameters when appropriate or kept with default parameters to suit the different types of analysis.

## Data availability

The Illumina reads Hiseq 2500 (2× 150 bp as well as the 10x Genomics data) were deposited at NCBI short archive under SRA accession PRJNA524026 (Biosample accession SAMN10995380). The RNA sequencing data as well as the Oxford Nanopore were deposited at NCBI short archive under BioProject (PRJNA600770, SRA ID SUB6799681). The final genome, is deposited at NCBI GeneBank (https://www.ncbi.nlm.nih.gov/genbank/wgs_update/) under accession (JABAOJ000000000), and supplementary data files, mitochondria genome as well as HGT validation are deposited at Dryad database (https://datadryad.org/stash/share/yyBU31Aj2_n2H9QYCPkG-hoZcYf63-2OToegnkmBcV8).

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

## Acknowledgements

We would like to thank all the personnel of the Khalifa center for Genetic Engineering and Biotechnology (KCGEB) who made this phase one of the project come true. A special thanks to Mr. Sajid at KCGEB for taking pictures of *R. ferrugineus*. This project was funded by the Khalifa Center for Genetic Engineering and Biotechnology (KCGEB) and by the New York University Abu Dhabi Research Institute (G-1205C and G-1205i).

## Author contributions

K.A., C.D. and K.M.H. designed the experiments. K.M.H., D.N. and N.S. performed the analysis. M.D., B.K. and A.M. collected the *R. ferrugineus* samples and extracted high molecular weight DNA. J.J.S. did the flow cytometry analysis. K.M.H., K.A. and C.D. wrote the manuscript.

## Competing interests

The authors declare they have no competing interests.
