## [Peer Review File · Communications Biology]

Reviewers' comments:

Reviewer #1 (Remarks to the Author):

Authors have sequenced and analyzed the genome of the red palm weevil pest (*Rhynchophorus ferrugineus*) insect. The genome was compared with other select insect species, and points of evolution, structural variation, and gene expansion and losses.

Manuscripts reads very well, flow of the information and illustrations are nice, and methods used for data analysis are up to date too. There are lot of useful observations supported by good data analysis present throughout manuscript.

However, I have some minor comments.

Line 283-284, I found this very interesting. I wonder what events could be related to sharp decline of a population from the middle to the end of Pleistocene. It would be interesting to relate this decline with physical or biological events occurred between this period and include in the discussion section.

Line 336, Though I am not expert in the experimental part, I feel that PCR results should be shown as a panel in Figure 6 since those are confirmatory results.

It would be good to add table to main results on the gene family, number of expanded copies, along with Pfam domains dominant in most of them.

It may deviate from the problem addressed in the manuscript but I feel that one or two paragraphs on the comparison of transcription factors, signaling and metabolic proteins will add more significance to the presented analysis.

There are lot of spelling mistakes in the manuscript including in figure legends.

Reviewer #2 (Remarks to the Author):

Review of “The genome of the red palm weevil pest (*Rhynchophorus ferrugineus*) reveals key gene families functioning at the plant-beetle interface” by Hazzouri et Al.

I had originally thought this would be a quick review of a genome note, but the authors are to be commended on an excellent high quality genome assembly, annotation and analysis.

First the Major problem. I cannot find the assembly at NCBI. The data availability seems to think it is OK to not put the assembly in NCBI but rather host it on a private website. This is unacceptable. It does not count if it is not in an INSDC database such as NCBI, ENA DDJB. If no one can find your assembly, we will have to redo it. Just putting the reads in the SRA is not good enough. You found the other genomes in NCBI for comparison – do you not want your genome to support comparative work in the future?

In the future I hope that the editor will not send manuscripts out for review unless the data is already in public databases pre-publication – but rather send them back immediately to fix this error. The paper is unpublishable until the data and assembly is in NCBI. A screen shot of the current NCBI search that found no results.:

End rant.

The rest of the manuscript has only minor issues, that can also be fixed easily.

The genome assembly looks good, although I think there could be some discussion of the duplication rate of BUSCO scores. Overall though Table SF7 nicely explains the difference in genome size.

The TE analysis looks good, and seems like a good explanation of the genome size.

The standard gene family evolution analysis of ORs, bHLH, GSTs, P450s, and GH genes is a well-chosen set and will be a good foundation for future control work. In general I liked this section a lot.

The history of the GH16 gene was also enjoyable.

I really enjoyed the ancestral and present day population size calculations. It would be lovely to have these for every genome. Fig 5. is great.

Overall this is a nice genome analysis, and after the data is publicly available – and some minor revisions/typos below are quickly addressed it can be accepted. I do think that because there are few beetle genomes, especially compared to the speciosity of the order, and because of the agricultural importance, the manuscript will be well cited, and appreciated in the field. I would normally put minor revision or accept, but because of the data not being in NCBI or ENA I am putting major revision.

Minor issues:

Lines 61 – 65 the small paragraph starting “Whole genome and transcriptome sequence (RNAseq) databases are very useful...” seems unnecessary and redundant with other text – I think it can be profitably removed.

I’m not sure what a “length-weighted mean molecule length” is. So I looked it up, and it is not clear to me why you would weight, or how you weighted. – I don’t think it is necessary – perhaps just report the n50 molecule length?

Line 96 – for the readers – is this a contig or scaffold N50?

Table 1. I am looking at table 1. Which assembly is final and is in the database? Please make it easier for the reader to know. Again it would be nice to have both contig and scaffold N50s.

Line 103. I am not sure what a pseudochromosome is? – I assume it is a notification to the reader that the chromosome sequence has not been tested against physical evidence and

should not be trusted. Perhaps for the reader you could define it. Alternatively this manuscript suggests all kinds of new terms:

“Precision nomenclature for the new genomics, Harris A Lewin, Jennifer A Marshall Graves, Oliver A Ryder, Alexander S Graphodatsky, Stephen J O'Brien. GigaScience, Volume 8, Issue 8, August 2019, giz086, <https://doi.org/10.1093/gigascience/giz086> “

Personally I think the new term is just confusing and I'm OK with just “chromosome” even if I am aware it is a fasta file and not a cell I am looking at down a microscope. – For this manuscript – probably best just to define what you mean.

Line 146, please add a period after 1,488 bp and before H. hampei.

Response to reviewers' comments:

Reviewer #1:

Authors have sequenced and analyzed the genome of the red palm weevil pest (*Rhynchophorus ferrugineus*) insect. The genome was compared with other select insect species, and points of evolution, structural variation, and gene expansion and losses. Manuscripts reads very well, flow of the information and illustrations are nice, and methods used for data analysis are up to date too. There are lots of useful observations supported by good data analysis present throughout manuscript.

However, I have some minor comments.

Line 283-284, I found this very interesting. I wonder what events could be related to sharp decline of a population from the middle to the end of Pleistocene. It would be interesting to relate this decline with physical or biological events occurred between this period and include in the discussion section.

We agree with the reviewer, we discussed this decline of population from the middle to the end of Pleistocene in a paragraph added to the discussion at line 484-490.

Line 336, though I am not expert in the experimental part, I feel that PCR results should be shown as a panel in Figure 6 since those are confirmatory results.

We agree with the reviewer: We moved a snap shot of PCR results from the supplementary information to Figure 6 and updated it.

It would be good to add table to main results on the gene family, number of expanded copies, along with Pfam domains dominant in most of them.

We agree with the reviewer's suggestion. We summarized the gene families and their Pfam domains with expansion/loss and included this information as Table 2.

It may deviate from the problem addressed in the manuscript but I feel that one or two paragraphs on the comparison of transcription factors, signaling and metabolic proteins will add more significance to the presented analysis.

Although it does deviate from the main problem addressed in the manuscript, we added a paragraph about the comparison of transcription factors and signaling and metabolic proteins at line 221-227 and 453-457, as well as included a supplementary Fig. 9 and supplementary Table 2 regarding that.

There are lots of spelling mistakes in the manuscript including in figure legends.

We proof read the manuscript and corrected the English spelling throughout the manuscript, including in the figure legends.

Reviewer #2:

I had originally thought this would be a quick review of a genome note, but the authors are to be commended on an excellent high quality genome assembly, annotation and analysis.

First the Major problem. I cannot find the assembly at NCBI. The data availability seems to think it is OK to not put the assembly in NCBI but rather host it on a private website.

This is unacceptable. It does not count if it is not in an INDS database such as NCBI, ENA DDJB. If no one can find your assembly, we will have to redo it. Just putting the reads in the SRA is not good enough. You found the other genomes in NCBI for comparison – do you not want your genome to support comparative work in the future? In the future I hope that the editor will not send manuscripts out for review unless the data is already in public databases pre-publication – but rather send them back immediately to fix this error. The paper is unpublishable until the data and assembly is in NCBI. A screen shot of the current NCBI search that found no results:

End rant.

We would like to acknowledge this terrible mistake and agree with the reviewer. The Short archive data hosted in NCBI is now released. We uploaded the genome to NCBI and it is under accession (JBAOJ000000000) and added that to the revised manuscript. We also uploaded the genome and supplementary data files to another public database (dryad) and provided the link in the revised manuscript.

The rest of the manuscript has only minor issues that can also be fixed easily. The genome assembly looks good, although I think there could be some discussion of the duplication rate of BUSCO scores. Overall though Table SF7 nicely explains the difference in genome size.

We are pleased that the reviewer finds the genome assembly good. We have discussed that it is a metric for good assembly. The differences in genome size are addressed in Supplementary Fig. 8. We highlighted this in the discussion.

The TE analysis looks good, and seems like a good explanation of the genome size. The standard gene family evolution analysis of ORs, bHLH, GSTs, P450s, and GH genes is a well chosen set and will be a good foundation for future control work. In general I liked this section a lot.

The history of the GH16 gene was also enjoyable.

I really enjoyed the ancestral and present day population size calculations. It would be lovely to have these for every genome. Fig 5. is great.

Overall this is a nice genome analysis, and after the data is publicly available – and some minor revisions/typos below are quickly addressed it can be accepted. I do think that because there are few beetle genomes, especially compared to the speciosity of the order, and because of the agricultural importance, the manuscript will be well cited, and appreciated in the field.

I would normally put minor revision or accept, but because of the data not being in NCBI or ENA, I am putting major revision.

We thank the reviewer for this very positive assessment and have now uploaded the sequence to NCBI.

Minor issues:

Lines 61 – 65 the small paragraph starting “Whole genome and transcriptome sequence (RNAseq) databases are very useful...” seems unnecessary and redundant with other text – I think it can be profitably removed.

We agree with the reviewer comment and we removed unnecessary and redundant text related to this section in the revised manuscript.

I'm not sure what a "length-weighted mean molecule length" is. So I looked it up, and it is not clear to me why you would weight, or how you weighted. – I don't think it is necessary – perhaps just report the n50 molecule length?

Although the "length-weighted mean molecule length" is a terminology used for long reads assembly, we agree with the reviewer that this was not clear and reported an N50 length instead. We moved this section in the Supplementary information-Supplementary methods.

Line 96 – for the readers – is this a contig or scaffold N50?

Table 1. I am looking at table 1. Which assembly is final and is in the database? Please make it easier for the reader to know. Again it would be nice to have both contig and scaffold N50s.

We agree with the reviewer comment and we have included N50 contigs when applicable in Table 1 in the revised manuscript. We added that to the Illumina assembly/10x at the contig level and to oxford nanopore assembly. For the other hybrid assembly procedures, we cannot include N50 contigs because it relies on merging of scaffolds and not contigs. The same issue exists in the synteny approach, where reference based assembly was performed.

Line 103. I am not sure what a pseudochromosome is? – I assume it is a notification to the reader that the chromosome sequence has not been tested against physical evidence and should not be trusted. Perhaps for the reader you could define it. Alternatively this manuscript suggests all kinds of new terms:

“Precision nomenclature for the new genomics, Harris A Lewin, Jennifer A Marshall Graves, Oliver A Ryder, Alexander S Graphodatsky, Stephen J O'Brien. GigaScience, Volume 8, Issue 8, August 2019, giz086, <https://doi.org/10.1093/gigascience/giz086> “ Personally I think the new term is just confusing and I'm OK with just “chromosome” even if I am aware it is a fasta file and not a cell I am looking at down a microscope. – For this manuscript – probably best just to define what you mean.

We agree with the reviewer comment. We define in line 140-142, that these pseudochromosomes are not oriented: To be able to orient them using linkage groups, we would need a genetic map and thus to perform genetic crosses. For clarity, we define and added a sentence at the end ‘.....and to generate full oriented chromosomes’.

Line 146, please add a period after 1,488 bp and before H. hampei.

We added a period.

REVIEWERS' COMMENTS:

Reviewer #1 (Remarks to the Author):

Revised manuscript reads well, however I suggested following changes,

#comment 1 Please remove "in insects" from the following sentence

"Transcription factors (TFs) play a vital role in controlling gene regulation and many diverse physiological processes in insects."

Same is for the sentence "The importance of TFs in orchestrating many physiological processes is vital to living organisms and especially to insets"

TFs orchestrate all physiological process (not many) in all living organisms. It is not special case for insects. Please revise such sentences everywhere in the manuscript.

#comment 2 Instead of "we think", I think it is better to start with, "which may" or "which might" like words in the following sentence.

"The mid to the end of Pleistocene experienced glacial-interglacial cycling and around 900 ka, there is evidence in the two southern Oceans of sluggish thermohaline overturn⁷⁶, which we think created isolated refugia that have receded the native Pacific and the India coconut (*Cocos nucifera*)."

Reviewer #2 (Remarks to the Author):

My earlier concerns about public access to the genome sequence have been fully resolved, and I can see the genome assembly in NCBI.

The minor issues have all been resolved.

I recommend the manuscript be accepted for publication, no further changes are needed.

REVIEWERS' COMMENTS:

Reviewer #1 (Remarks to the Author):

Revised manuscript reads well, however I suggested following changes,

#comment 1 Please remove “in insects” from the following sentence

“Transcription factors (TFs) play a vital role in controlling gene regulation and many diverse physiological processes in insects.”

Same is for the sentence “The importance of TFs in orchestrating many physiological processes is vital to living organisms and especially to insets”

TFs orchestrate all physiological process (not many) in all living organisms. It is not special case for insects. Please revise such sentences everywhere in the manuscript.

We agree with the reviewer comment and fixed that and revise that all over the manuscript.

#comment 2 Instead of “we think”, I think it is better to start with, “which may” or “which might” like words in the following sentence.

“The mid to the end of Pleistocene experienced glacial-interglacial cycling and around 900 ka, there is evidence in the two southern Oceans of sluggish thermohaline overturn⁷⁶, which we think created isolated refugia that have receded the native Pacific and the India coconut (*Cocos nucifera*).”

We agree with the reviewer comment and we fixed that and used “which might” in the following sentence.

Reviewer #2 (Remarks to the Author):

My earlier concerns about public access to the genome sequence have been fully resolved, and I can see the genome assembly in NCBI.

The minor issues have all been resolved.

I recommend the manuscript be accepted for publication, no further changes are needed.